# Particulate Matter Emission Factors for Dairy Facilities and Cattle Feedlots during Summertime in Texas

**DOI:** 10.3390/ijerph192114090

**Published:** 2022-10-28

**Authors:** Mohammad Ruzlan Habib, El Jirie N. Baticados, Sergio C. Capareda

**Affiliations:** Biological and Agricultural Engineering Department, Texas A&M University, College Station, TX 77843, USA

**Keywords:** summer PM emission, PM ratio, AERMOD, drone, low-cost sampler collocation

## Abstract

Particulate matter (PM) emissions from dairies and feedlot sources require regular emission factor update. Likewise, development of simple measurement technique to accurately measure pollution concentration is warranted to limit the impact of air pollution and take necessary actions. During June of 2020, a dairy facility from central Texas and a feedlot from the Texas Panhandle region, titled as Dairy B and Feedlot C, respectively, were chosen for measurement of PM emissions in the state of Texas to represent dairy facilities and cattle feedlots PM emission rates. Four stations, each assigned with an EPA-approved Federal Reference Method (FRM) sampler, Texas A&M University (TAMU) designed sampler and handheld non-FRM AEROCET (MET One Instruments) sampler for collocation, were selected within each sampling locations. Drones were also utilized mounted with a handheld AEROCET sampler for simultaneously sampling at a certain height. PM_2.5_ emissions of Dairy B were all below 24-h PM_2.5_ standard of 35 μg m^−3^ as specified by National Ambient Air Quality Standards (NAAQS) even at the 98th percentile. The PM ratio between regulated PM_10_ to PM_2.5_ was determined to make an estimate of relative percentage of coarser particles to fine particles in both feedlot and the dairy representative animal facilities. The maximum mean emission factor determined using AERMOD for PM_2.5_ and PM_10_ was found to be 0.53 and 7.09 kg 1000-hd^−1^ d^−1^, respectively, for the dairy facility while 8.93 and 33.42 kg 1000-hd^−1^ d^−1^, respectively, for the feedlot. A conversion factor and correlation matrix were developed in this study to relate non-FRM sampler data from the handheld AERCET samplers with FRM samplers. Cheaper handheld samplers (AEROCETs) may play a potential role in quick and relatively instant measurement of PM emissions to initiate necessary preventive actions to control PM emission from dairy facility and feedlot sources.

## 1. Introduction

Due to the extensive meat demand of the increasing world population, cattle production in concentrated animal feeding operation (CAFO) systems has grown trendily larger, resulting in a prominent environmental issue of particulate matter (PM) emissions from the dairies and feedlots. As of 2019, the US inventory for agricultural counts estimates over 94.5 million cattle heads, with about a 7% increase from 2015, including 41 million cow heads of which Texas accounts for 5.2 million cattle heads [1]. As feedlot is the final stage of cattle production, providing a confined area for feeding, there are 26,586 feedlots in the US where a cow stay time or feeding period ranges from 90 to 300 days depending on desired weight and grade [2,3]. During this cattle production period, animal movement in the CAFO system and defecation of manure waste creates air pollution. The pollution is from the emission of dust, volatile organic compounds (VOC), greenhouse gases (GHGs) (methane, nitrous oxides, and carbon dioxides), etc. Similarly, dairies containing a large number of animal heads can significantly pollute the air, of which particulate matter holds the majority in mass. Depending on the body size and whether the cow is milking or not, cattle can generate around 5–7% manure of their body weight, which is about 5.5 kg of dry mass from slurry per animal per day [4]. It is important to note that dairy animals excrete higher manure contributing to more PM emissions than the beef cattle [4]. Although genetic selection of cows can contribute to limiting CH_4_ emissions in dairy [5], no known studies have been conducted on the impact of breeding on direct PM emissions. However, the manure from cattle and the soil create a mixed sub-surface in feedlots. This is pulverized into dust by animal activities ready to be dispersed into the atmosphere by blowing wind.

Particulate emission mostly includes the total suspended particles (TSP) with an aerodynamic diameter of up to 25–45 μm (mostly), PM_10_ with an aerodynamic diameter of 10 μm or less, and PM_2.5_ with an aerodynamic diameter of 2.5 μm or less [6,7]. Adverse health effects can be rendered by these particles, especially with the finer sizes that can be inhaled directly into the alveoli of the lung [8]. Along with direct inhalation, a major indirect health impact of PM emissions is that they can carry odorous gases leading to exposure to the atmosphere and humans; eventually causing lung issues, cardiac arrhythmia, pneumonia, heart attacks, etc. [9,10,11]. Additionally, the emissions of inhalable PMs containing badly odorous gases from the feedlot sources can be transported by the wind to nearby human-living areas [12,13,14]. Concurrently, the emissions carried to nearby roads may reduce visibility, especially in the early-mid-evening period [15].

The emission factor (EF) provides a basic footprint that can be correlated to the quantity of air pollutants exerted in the atmosphere. This has long been used as a sole strategy for air quality regulatory purposes and control as it represents the emission status for a large variety of sources. PM emissions from feedlot sources require regular EF update and simple measurement techniques to evaluate air pollution may be required to take immediate and necessary actions. Currently, the National Ambient Air Quality Standards (NAAQS) proposed annual average primary standards for PM_2.5_ and PM_10_ are 12 μg m^−3^ and 150 μg m^−3^, respectively [6]. Texas, Nebraska, and Kansas together hold about 65% of the total cattle heads fed in a feedlot in the United States. Texas leads the states with about 23.4% of total cattle heads fed in the US feedlots. Since feedlots are sources of huge dust emissions, Texas feedlots require regular emission factor update and take control measures to avoid regulatory actions [3]. However, the PM emission factor data from Texas CAFOs have been outdated for more than a decade. Due to the increased size and number of cattle production establishments in the past few years, PM emissions are anticipated to change substantially. This will require updating the emission factor data inexorably in the feedlots with more scrutiny based on actual field experiments. Added to this, most of the previous sampling and data collection was based on conventional sampling techniques that require hourly filter changing and continuous in-person monitoring. Thus, an actual field sampling with simple and quick sampling tools is warranted to measure the emission factors in dairies and feedlots. Additionally, since there are rapid dynamic changes in the emissions, real-time measurement of emission factor at higher time resolution using different sensor-based technologies can be beneficial in controlling the emissions.

Net particulate emissions, as well as emission factor, vary depending on numerous factors, encompassing dairy or feedlot size, animal heads, manure amount, weather conditions, soil status, human activities, etc. However, some of these factors are considered while calculating emission factors for dairies and feedlots by two most familiar techniques named the inverse dispersion modeling [16] and the flux gradient technique [17]. Although there have been limited studies on cattle feedlot emission factor calculation in the United States, most of the studies reported the inverse dispersion modeling approach as an efficient technique for determining the emission rates and therefore the emission factors. As the emission factors fluctuate seasonally and regionally, one Texas cattle feedlot reportedly measured 71 kg 1000-hd^−1^ d^−1^ without rainfall events and 5.4 kg 1000-hd^−1^ d^−1^ with the rainfall events for the PM_10_ emission factor [18]. Another study in Kansas reported an emission factor of 57, 21, and 11 kg 1000-hd^−1^ d^−1^ for TSP, PM_10_, and PM_2.5_, respectively, for a sampling campaign of specific days over two years [17]. As there are limited studies conducted on PM_2.5_ emission factor determination, it is imperative to measure and update PM_2.5_ emission factors for commercial beef cattle feedlots in the United States.

Even though there have been some recent advancements in dust emission measuring techniques, a collocation of more simple tools with the US EPA designated Federal Reference Method (FRM) is required for standardizing simple and quick PM emission data. A Federal Reference Method is specified as a reference method for sampling and analyzing ambient air for an air pollutant based on national primary ambient air quality standards for PM_10_ and PM_2.5_ in Title 40 of the Code of Federal Regulations (CFR) [19]. Following the FRM method, instantaneous data on the time-resolved size distribution of PM_10_ and PM_2.5_ and the real-time concentrations will help take necessary actions sooner for emission abatement in the feedlots.

The hypothesis of the study assumes that the inverse modeling approach can be utilized for the determination of emission factors in large commercial dairies and feedlots. Likewise, the use of small handheld instruments can be collocated for simple and quick interpretation of PM emission status. Hence, the design of this research additionally focuses on the comparison of the dust emission data from the quick samplers with the FRM samplers. A subsequent step of updating the revised emission factors and concentrations of PM emissions will follow as well.

The specific objectives include:To provide an update of the PM_2.5_, PM_10_, and TSP emission factors for dairies and feedlots,To develop a quick and simple emission measuring tool in the animal facilities to hasten effective abatement procedures, andTo collocate the FRM samplers with the non-FRM samplers to standardize the simple and low-cost dust emission measuring tools.

## 2. Materials and Methods

The cattle population in Texas is mostly centered in the Panhandle known as the Northern High Plains region with 250,000 to 499,000 cattle head ranges and in the central Texas region. This project thus identified a representative dairy facility located in central Texas (with a cattle average population of 450 to 500) and a cattle feedlot facility in the Panhandle of Texas (with a cattle population of 45,000 to 60,000) for the measurement of PM emissions. Both represent the majority of dairy and cattle farms in the state.

### 2.1. Sampling Location

During June of 2020, the dairy facility and the feedlot, titled Dairy B from the central Texas and Feedlot C from the Texas Panhandle region, respectively, were chosen for measurement of PM emission in Texas dairy facilities and cattle feedlots. The areas of Dairy B and feedlot C are 15 acres (approximately) including a pen area of 1.15 acres and 500 acres including a pen area of 220 acres, respectively. The average length of stay for the cattle in the feedlot is about 180–190 days depending on the resultant carcass amount and quality. Though the predominant hourly wind direction in the sampling sites varies throughout the year, the direction is most often from the south during the summer. Additionally, the average wind speed in the Dairy B site ranges from 3.3 to 4.7 m s^−1^ while in feedlot C ranges from 4.6 to 5.9 m s^−1^ during the summer. The temperature, wind speed, and relative humidity were recorded during the sampling work.

### 2.2. Experimental Design

One upwind and three downwind (collocated) stations were selected within each sampling location for measuring the dust emissions in the dairy facility and the feedlot throughout a week (Appendix A). Each station was assigned with a manually operated TAMU (Texas A&M University) designed sampler for TSP and PM_10_ designed by a previous study [20], and a handheld battery-operated mass profiler (model: AEROCET 831 from Met One Instruments Inc.; Grants Pass, OR, USA). Three FRM PM_2.5_ samplers were used for 3 downwind stations. The programmable FRM samplers were of basically three types equipped with PM_10_ sampling inlet and a Very Sharp Cut Cylinder (VSCC) for PM_2.5_ collection from BGI Inc.; Waltham, MA, USA: (i) model BAM 1020 particulate monitor, Met One Instruments Inc.; Grants Pass, OR, USA, (ii) model PQ-200 air sampler, Mesa Laboratories Inc.; Butler, NJ, USA, (iii) model E-FRM 120, Met One Instruments Inc.; Grants Pass, OR, USA. Although BAM is considered as a Federal Equivalent Method sampler, it will be called FRM sampler collectively with other FRM samplers in the present study. Stations 1, 2, and 3 were set up with BAM, PQ-200, and E-FRM sampler, respectively. Horizontally, Station 2 and station 3 were 135 m far from station 1 to the east and west, respectively. The horizontal gap between the samplers in each station were within 1–3 m. Additionally, the samplers were placed side by side maintaining the measurement height of around 1.5–2 m from the ground at each station. The upwind station PM_2.5_ was measured with TAMU designed sampler equipped with a PM_10_ sampling inlet and a VSCC. The ground AEROCET’s were placed side-by-side with the FRM samplers and the BAM sampler maintaining a height of around 1.5 m. The AEROCET was used to simultaneously monitor the PM_2.5_, PM_10_, and TSP levels for collocation with the data from manually operated samplers and FRM samplers.

Additionally, the study used small lightweight designed Unmanned Aerial Vehicles (UAVs or drones) to collect PM emissions from the sampling stations. The drone model was DJI Matrice 600 Pro equipped with an A3 Pro flight controller (DJI, Shenzhen, China). The drones were mounted with an AEROCET sampler and a long sampling tube head was connected to it to avoid air turbulence from the drone blades. Featured with GPS positioning and accurate pressure altitude, the drones had a flight time of up to 30 min with fully charged batteries. After every 30 min, another drone replaced the drone that has completed the sampling episode. The overall sampling process followed a previous air quality sampling work for PM emissions [21,22,23].

All the samplers were calibrated before deployment for sampling at a flow rate of 16.7 L min^−1^ and audited for leak test. PM_2.5_ concentration was measured using the FRM samplers operated at a constant flow rate of 16.7 L min^−1^ (accuracy ±0.50%). The AEROCET samplers were also calibrated and flow rate tested at 2.83 L min^−1^ (accuracy ±5.0%) before sampling. This difference in flow rate is a major limitation for the AEROCET samplers as the drawing speed can interfere the receiving of the particles from the air. Added to this, AEROCET samplers are designed specifically for indoor environment and hence the PM concentration measured by these can be hugely impacted by outdoor wind and rough temperature. However, the AEROCET samplers can run for about 8 h when fully charged, while the BAM needs continuous power source. For the drone samplers, it requires charging every 30 min for continuous operation. Additionally, BAM has long storage capacity of 182 days, while the AEROCET samplers have about 2 days.

### 2.3. Emission Calculations

The non-programmable TSP and PM_10_ samplers pressure changes were recorded every hour and flow rate was monitored throughout the sampling duration. The pressure drops or rise was later used to verify the flow rate of the samplers. The Whatman filter preparation, pre-weighing, and post-weighing were carried on following the SOP for PM Gravimetric Analysis [24]. A continuous filter tape roll from the Met One Instruments Inc.; Grants Pass, OR, USA was used for the programmable FRM sampler BAM1020 model. The non-FRM PM_10_ and TSP samplers contained the Whatman filters and the AEROCET samplers required no filters. The FRM samplers were run continuously for 24 h while the non-FRM samplers for TSP and PM_10_ measurement were run during the daytime from 8 a.m. to 6 p.m. as they require manual filter replacement and monitoring. The sampling schedule is presented in Table 1. The AEROCETs were run for 24 h throughout the sampling period equipped with charging backup.

Air quality dispersion modeling has become a prevalent technique to predict pollutant concentrations emitted by a source at selected downwind receptor locations using mathematical formulations and meteorological inputs. The American Meteorological Society/Environmental Protection Agency Regulatory Model (AERMOD) by BREEZE AERMOD from Trinity Consultants (Charlotte, NC, USA) was used for dispersion modeling of this sampling work to estimate the PM emissions. Surface parameters and meteorological parameters were estimated considering the US EPA guidance for AERMOD [25]. It is worth noting that AERMOD is based on Gaussian plume model and uses Gaussian distribution for stable conditions to measure vertical or horizontal distributions [26]. However, for unstable conditions, a bi-Gaussian concept has been proven more acceptable for vertical concentration distribution [27] while a Gaussian model is still usable for horizontal distributions. Detailed description regarding the AERMOD principles is explained in previous studies [26,27]. The overall modeling was followed in three steps: (i) AERMET running, a software tool for meteorological data preprocessing, (ii) terrain data processing for AERMOD, and (iii) AERMOD running for dispersion modeling with the 3D analyst for visualizing the dispersion. AERMET processing requires collecting the surface raw data from NOAA and upper air or radiosonde data from the nearby certified weather stations, and onsite data collected from the sampling location. A quality assurance step was followed to audit and input any missing data of various meteorological parameters in the AERMET. Terrain data were collected from the US Geological Survey (USGS) and processed for AERMOD use. Discrete receptors were used at the sampler height in the AERMOD for distributed emission prediction. The flagpole option from the receptor was utilized for the drone samplers EF determination as it had a vertical height difference (10 m from the ground) from the other samplers or local terrain. Added to this, the area sources option was used in the AERMOD run for distributed emissions. The AERMOD was run utilizing the outputs from the AERMET and terrain data. As default, the AERMOD model handles the missing data processing routines; however, it gives a warning if the data are missing for more than 10%. Surface characteristics can impact the meteorological conditions through radiative (i.e., albedo) and non-radiative (i.e., Bowen ratio and surface roughness) properties and thus used in the model (Appendix A). Particle size distribution (PSD) and scanning electron microscopic (SEM) image were also analyzed for sampling location soil characterization (Appendix A).

The PM concentration from an animal facility can be varied throughout the day at different stations due to the variation of wind speed, wind direction, humidity, unpaved surface, and animal movement at different sections of the facility. This will additionally impact the emission factor estimation at different stations every hour. Hence, the following assumptions that were considered: (i) For 1-hr averaging time, emission flux were constant throughout the pen surface; (ii) the emitted particles pathway was a function of a wind direction only; (iii) both the animal facilities had flat terrain and area sourced pen surface; and (iv) particle removal mechanism was based on dry depletion only. It is also important to note that the BAM sampler draws air at 16.7 L min^−1^ and the AEROCET sampler draws at 2.83 L min^−1^.

To indicate the actual airflow rate into the designed TAMU sampler’s inlet head, a magnehelic gauge (Dwyer Instruments, Michigan City, IN, USA) measuring pressure drop across the sharp-edge orifice was used and connected with a differential pressure transducer (PX274, Omega Engineering, Inc., Stamford, CT, USA). A diaphragm pump of 0.09 kW was used to draw the air at 16.67 L min^−1^ flow rate. During the sampler runs, a data logger (HOBO H8 RH/Temp/2x External, Onset Computer Corp, Pocasset, MA, USA) was used to record the relative humidity and temperature for air density correction analysis. The general PM concentrations collected by all samplers except BAM1020 and AEROCETs were calculated using Equation (1).
(1)$$Concentration\;\left(\frac{mg}{m^{3}}\right)=\frac{Filter\;Postweigh\left(\mu g\right)-Filter\;Preweigh\left(\mu g\right)}{Flow\;Rate\;\left(\frac{L}{\min}\right)\times Sampling\;Time\;\left(\min\right)}$$
where the standard flow rate was maintained at 16.67 L min^−1^, so that an hourly (60 min) sampling period gives a total 1000 L or 1 m^3^ of air volume to estimate the emission concentration in mg m^−3^ or converted to μg m^−3^.

Data screening was conducted considering the unexpected high peaks and outliers that fall between 3 to 4 standard deviations above the mean of raw data that may occur due to fugitive dust sources. Other important screening processes utilized include whether the data follow the proper order of sizes, checking the data gap, periodicity of the peaks, and excluding continuously same values for a time period.

### 2.4. Emission Factor (EF) Calculation

A basic difference in the calculation process of emission flux and emission factor is that the emission flux is determined as per area of the pollution source while emission factor basically uses the animal head that generates the dust emission. For the back calculation in AERMOD, Equations (2) and (3) were used [28]. The PM concentrations were measured individually in each setup point and EF was developed based on that. For the emission flux measurement, the reverse calculation approach was followed, where a predicted emission flux was utilized for modeling to get the distributed PM concentration from the model at a vertical height of 1.5–2 m from the ground at the stations.
(2)$$EF=\frac{Q_{yr}\times A}{AU}$$
where *EF* is the Emission Factor (kg hd^−1^ d^−1^), *AU* is the number of animal unit (hd), A is pen surface area (m^2^), and *Q_yr_* is the mean annual emission flux in μg m^−2^ day^−1^ and measured using Equation (3).
(3)$$Q_{o}=\frac{Q_{A}}{C_{A}}\times C_{o}$$
where *Q_A_* is the assumed emission flux of 1 μg m^−2^ s^−1^ for modeling, *C_A_* is the model predicted net 1-hr concentration (μg m^−3^) that is primarily assumed and then verified through back calculation from the model output concentration, and *C_o_* is the measured net 1-hr concentration in (μg m^−3^).

### 2.5. Statistical Analysis

The statistical significance level was considered when *p* > 0.05 in all cases. The collocated data from AEROCETs were compared with the FRM and non-FRM samplers for PM_2.5_, PM_10_, and TSP emission measurement. A paired sample *t*-test was carried out among the samplers to evaluate the proportions of the achieved values. In addition, a correlation analysis was conducted to measure the relationships in concentrations among the samplers using SPSS and Microsoft Excel (Version 2020).

## 3. Results and Discussion

### 3.1. Meteorological Data

The weather conditions data considering wind speed and direction, relative humidity, and temperature of the Dairy B and Feedlot C were plotted using AERMOD and presented in Figure 1. The majority wind speed in Dairy B ranged from 2.6 m s^−1^ to 7.8 m s^−1^ for 71.7% of the time and wind direction was mostly from the South and South Southeast (120° < θ < 180°) for about 47.8% of the time. The mean relative humidity and temperature in the dairy facility were 69.2% and 30.2 °C during the sampling week. Feedlot C had the predominant wind direction from the south, more specifically at an angle of between 120° < θ < 240° of north, for about 83.7% of the time. The direction was reflected in the onsite data during the sampling time as well. The majority of the wind speed was around 7.50 m s^−1^ for 44.4% of the data with an average wind speed of 6.3 m s^−1^ while the average temperature was 26.3 °C. Alongside hoofing or cattle activities, meteorological conditions play a stellar role in PM formation, size distribution, transportation, and settling [7,29]. The impact of precipitation and wind speed and direction on the PM emission and dispersion was also previously studied elsewhere [30]. 

### 3.2. PM Concentration

The daily 1-h average for PM emissions collected at both sampling sites for each day is presented in Figure 2. All TSP, PM_10_, and PM_2.5_ concentrations in Feedlot C were considerably higher than in Dairy B. The TSP, PM_10_, and PM_2.5_ in the feedlot followed in respective order in all stations each day. Among the sampling days for Feedlot C, the highest hourly average TSP, PM_10_, and PM_2.5_ were collected on June 16 with the value of 4734, 2597, and 100 μg m^−3^, respectively. The maximum hourly average PM_2.5_ of 200.37 μg m^−3^ was measured on June 15 at station 03 of Feedlot C, while the minimum concentration of 1-hr average for PM_2.5_ of 5.13 μg m^−3^ was measured on June 18 at the upwind station. Except for this value, on all the other days, the 1-hr average PM_2.5_ concentration measure for 10 h (8 a.m. to 6 p.m.) exceeded the NAAQS specified annual average primary standard for PM_2.5_ of 12 μg m^−3^. Except on June 15, the upwind station’s PM_10_ value, all days’ hourly average PM_10_ concentration exceeded the NAAQS specified 24-h average primary standard for PM_10_ of 150 μg m^−3^. The day-to-day PM concentrations varied probably due to the day-to-day weather differences and animal activities. Compared to this study, 3 feedlots with large animal heads of Texas High Plains and west central Texas measured lower concentrations of TSP and PM_10_ of 700 and 285 μg m^−3^, respectively [29]. Supporting the current research, another study in a feedlot in Kansas revealed the 24-hr concentration to be exceeding the NAAQS standard for 69% of the samples in 2007 indicating that 1-hr concentrations can exceed the NAAQS specified standards [28]. However, a 24-h sampling period is necessary to better compare the concentrations with the NAAQS standard. As the current study measured PM concentration for one week in the summer season only, it may not provide the best comparison to the NAAQS standard. Moreover, summer emission rates are commonly much higher for prolonged dry periods of pen surface moisture [28]. For this ground, considering the summer sampling only, annual emission was found higher in both dairy and feedlot than the NAAQS standard except for the PM_2.5_ concentration from the dairy.

The upwind PM_2.5_ concentrations of Feedlot C were compared similarly to two feedlots of Kansas with 25,000 and 30,000 heads capacity [28]. Dairy B PM concentrations were not very consistent such as that in Feedlot C mostly due to the lack of enough hourly emission data. Station 1 of Dairy B, placed near the loafing pen, had the highest 1-hr concentration of PM_2.5_ on 9 June while the lowest concentration was collected on 5 June. As cattle heads and area were much lower in the dairy than the feedlot, the overall TSP, PM_10_, and PM_2.5_ were lower in the dairy than the feedlot although sampling site temperature, relative humidity, and wind speed and direction play a major role in this regard. 

PM_2.5_ concentrations of Dairy B were all below the 24-h PM_2.5_ standard of 35 μg m^−3^ specified by NAAQS even at the 98th percentile of the concentration. The highest daily 98th percentile values for Dairy B were measured at 24 μg m^−3^ during the sampling period. On the other hand, Feedlot C PM_2.5_ concentrations exceeded the 24-h NAAQS standard as well as the annual NAAQS standard. The dairy facility upwind station mean PM concentrations had reasonably higher concentrations than the downwind stations as 15.1% of the time, the wind blew to the opposite direction of the regular wind or probably due to noticeable vehicle movement near the upwind station. Added to this, the downwind station 2 was about 114 m away from the dairy barn to the north. In addition, the feedlot daily downwind concentrations were 3 to 6 times higher than the upwind stations even though there was a roadway close (25 m) to the upwind station. However, this condition is quite common in these animal facilities.

From the diurnal PM concentration, the mean PM_2.5_ in the BAM sampler was measured 699.5 μg m^−3^ with the value of 985 μg m^−3^ at 11 p.m. for Station 1 of Feedlot C (Figure 3), which ranged from 468.5 to 935.5 μg m^−3^ at 95% confidence interval. To illustrate the maximum possible PM emission concentration diurnally, the PM_2.5_ values of 985 μg m^−3^ from BAM were taken into account while constructing Figure 3 which was screened while constructing Figure 2. However, please note that the BAM sampler is designed to measure a maximum of 985 μg m^−3^ of PM_2.5_ data. Wind gusts or tiny dust tornadoes that were observed during the sampling period in the feedlot surroundings can be the possible reasons behind this significant concentration. Added to this, cattle activity during the summer is expectedly higher than in the winter season for more comfortable weather conditions, specifically at nighttime. Hence, in the BAM sampler, higher PM concentration was recorded from the late night to midnight in the feedlot. Simultaneously, increased feeding conversion from a favorable environment may impact manure production as well as the emission and formation of particulate matter. The inset of Figure 3 shows the mean PM_2.5_ during the collocated sampling times from 8 a.m. to 6 p.m. Nonetheless, it was observed that the average TSP and PM_10_ concentration was higher during early morning, and then a peak was observed at around 3 p.m. It should be mentioned that station 01 in feedlot C was 25 m away from the nearest pen surface. The diurnal variations in the downwind station’s PM concentration were previously explained resulting from time-varying emission rates by cattle activity and real-time changes in the atmospheric conditions [31].

### 3.3. Ratio of PM Emissions

For PM ratio determination, screening was conducted in which zero to negative values and outliers or extra-ordinary high differences were sorted out from the collected PM concentration’s raw data. After screening the feedlot data, 74.5% of the data for the PM_2.5_/PM_10_ ratio, 72.5% of the data for the PM_10_/TSP ratio, and 61.7% of the data for the PM_2.5_/TSP were plotted for frequency distribution (Figure 4). More than 80% of the PM_2.5_/PM_10_ and PM2.5/TSP ratio was found below 0.2 compared to around 45% of the ratio found in a Kansas feedlot [17]. At the dairy facility, PM_2.5_ concentration was undetected by TAMU-designed samplers for a few hours in some stations. Hence, AEROCET data were utilized to generate the PM ratio for missing hours. However, a similar pattern was also observed for PM_2.5_/PM_10_ and PM_10_/TSP distribution for the dairy facility in the same way as the feedlot location. Furthermore, PM_10_ values for the dairy were found closer to TSP for 60% of the values. The current study means of the PM_10_/TSP ratio of around 0.51 for the feedlot were slightly higher than a study in a feedlot in Texas [29]. More than 64% and 67% of the emissions data were below 0.2 for PM_2.5_/PM_10_ and PM_2.5_/TSP ratios, respectively. Considering this spectrum and the above discussion, overall particles emitted in the dairy and feedlot were coarser which accords with a previous study [7]. The PM ratio in a dairy facility and feedlot also illustrates the impact of the weather and manure production on PM formation. Although submicron aerosol particles increase in size mostly by adhesion, coagulation, and condensation, the size change of particulate matter in the atmosphere by high wind gradient is somewhat unknown. 

### 3.4. Collocation of AEROCETs and Drone AEROCETs with BAM

The drones were mounted with AEROCET samplers and hence could measure all PM emissions data. Using this, a collocation of comparative data was analyzed with US EPA-approved BAM samplers PM_2.5_ value in downwind Station 1 for both Feedlot C and Dairy B (Figure 5). From the collocation in Feedlot C, it was observed that the ground AEROCET measured PM_2.5_ concentration was lower than the BAM measured concentrations followed by the Drone AEROCET measured concentrations. Notwithstanding, a pattern of relationship was observed between the ground AEROCETs and the continuous BAM Sampler, which in turn may provide useful information to get a quick and instant interpretation of the actual PM concentrations. Since the drones were flown 10 m above the ground, there will be a difference in PM concentration with the ground AEROCET and the BAM sampler due to the gravitational settling of the particles. This was also noticeable in the collocation study. However, this significant difference of drone AEROCET with BAM sampler may also occur from drone air turbulence from its blade changing particle’s path away from the inlet. For Dairy B, the correlation of the collocated samplers was fairly well understood. Though there was a dissonant correlation between the handheld samplers and the BAM sampler in the dairy, the ground AEROCET and the drone AEROCET were quite related. The collocated samplers in the feedlot were moderately associated exhibiting a similar pattern to the BAM, especially the ground AEROCET sampler PM_2.5_ (Figure 6d). The collocation observation may indicate the possible use of AEROCET samplers as a quick measurement technique considering some factors calculated with the AEROCET sampler to get the actual US EPA-approved sampler’s emissions. The sampling event had no rainfall occurrence observed in both locations. Added to this, the impact of relative humidity on hourly emissions were correlated for both the feedlot (Figure 6d). There are some noticeable peaks observed especially from midnight to early morning expressing a similar opinion to emission distribution [32] while the relative humidity during those peaks was correspondingly low. In some cases, the RH (%) correlation was observed in the next emission hours for PM_2.5_ concentration. Additionally, a major observation is to take into account as a limitation that the measurement range for the BAM sampler was limited to 985 μg m^−3^ and the AEROCET sampler was limited to 1000 μg m^−3^, above which the concentrations could be erroneous. Keeping this in mind, the PM concentrations would probably be the best fit to compare the AEROCET values that are below the 1000 μg m^−3^ limit. In this case, the PM_2.5_ values were kept below the limit for this summer sampling. However, to overcome the outdoor complexities for a low-cost instrument such as AEROCET, regression and statistical correlation analysis is vital.

All the BAM, AEROCET, and drone AEROCET data showed positive skewness in the correlation analysis by SPSS while the mean and median are higher than the mode of the dataset. The ground AEROCET’s PM_2.5_ had a higher mean and standard deviation with *n* = 89 samples than the drone mounted AEROCET’s, with *n* = 39 samples. This also elucidates that the upper atmosphere may have lower particles of high aerodynamic size PMs due to fast settling. It is also expected that submicron particles settling times are much higher than heavier PMs. The Pearson correlation indicates the ground AEROCETs to have a positive correlation with the BAM sampler (Appendix A). As the AEROCETs mounted on the drones are negatively correlated with the BAM sampler, it can be extrapolated that the increase of the BAM sampler value does not increase the drone AEROCET’s value proportionally. The correlation equation for BAM with AEROCET and drone AEROCETs are y = 0.5321x + 12.276 and y = 0.2267x + 2.8522, respectively (Figure 6). In addition to this, the conversion factor (CF), the ratio of USEPA approved FRM sampler (BAM) PM_2.5_ emission data to non-FRM samplers AEROCET PM_2.5_ and drone AEROCET PM_2.5_ emissions data were 1.45 and 3.61, respectively. The conversion factor was upheld by a paired sampled *t*-test for means with *p* < 0.05 between the BAM sampler and low-cost samplers. The paired sample *t*-test was conducted with *n* = 89 samples for the AEROCET samplers and *n* = 39 samples for the drone AEROCET samplers. However, rather than the linear regression, a polynomial regression taking wind speed into account may improve the FRM sampler and AEROCET relationship as the wind can be of major concern for low-cost sensor samplers [33]. 

### 3.5. Emission Factor

Emission factors aid in understanding the PM emissions from a source under different conditions and were generated with the help of AERMOD dispersion modeling. The emission flux, a precursor for dispersion modeling, assumes a certain value and gives PM concentration that accounts for the weather conditions in the modeling. Dairy B station 1 measured a maximum mean EF of 0.70 kg 1000-hd^−1^ d^−1^ for PM_2.5_ on June 9 which was found to be 2- to 5-fold less than a study conducted in a ventilated free-stall barn for two years [34]. There are no standards for emission factor by US EPA yet and very limited studies have been conducted on free-stall dairy barn’s PM_2.5_ emission factor. The maximum mean PM_2.5_ emission factor was 8.55 kg 1000-hd^−1^ d^−1^ on June 6 considering a 24-h sampling period (Table 2). A high standard deviation was noticed in the estimated emission factors. The PM concentrations due to positive skewness in the dataset may also indicate that the concentration may have been overestimated. This also attributes to the diurnal variability of the PM emission factors in open feedlot [35] for only a weeklong sampling period. Perhaps the annual emission data may reduce the non-normal distribution of the dataset. Nonetheless, compared to PM_2.5_ EF of 12 kg 1000-hd^−1^ d^−1^ estimated in California feedlot during the dry season [36], the present study PM_2.5_ emission was found nearly 2-fold lower (Table 3). Additionally, it is noteworthy that the used model with the inferred emission rate of 1 μg m^−2^ s^−2^ estimated values were fairly well correlated with the actual measured values at the same receptor coordinates. This was tested by establishing a correlation using a random sample day of 1-hr PM_2.5_ concentration data measured and the data from the model (Appendix A). As the pollution dispersion varied with the wind direction, dispersion modeling on AERMOD showed that the emission dispersion is mostly to the northeast during the sampling week at the dairy and feedlot facility (Appendix A).

An overall and station-wise PM_2.5_, PM_10_, and TSP emission factor estimated for the dairy is presented in Table 3. The current study was compared to a Texas dairy facility with PM_10_ EF of 4.7 kg 1000-hd^−1^ d^−1^ for a 24-h sampling in the free-stall portion [37]; feasibly, a 24-h sampling of the current study would fit more to the comparison study. Compared to this study, the annualized PM_10_ EF was estimated as 6.8 kg 1000-hd^−1^ d^−1^ for the southern High Plains of Texas, which contrasted the correction for AP-42 PM_10_ emission factor error [38]. A study by EPA on PM_10_ EF from free-stall dairies measured higher PM_10_ EF of 21 kg 1000-hd^−1^ d^−1^ than the present study [36], while a model dry feedlot measured PM_10_ of EF 15 kg 1000-hd^−1^ d^−1^ [39]. The overall EF for TSP of the feedlot, considering only summer sampling, was compared with the TSP EF of 81 kg 1000-hd^−1^ d^−1^ estimated in a Texas feedlot of moist pens using the ISCT3 dispersion model [40]. 

The dispersion of particulate matters from the free-stall dairy barns is impacted due to the barn shed as a wind barrier. In contrast, the open feedlot, comprised of the high open pen area and cattle activity, principally emits higher PMs than the free stall barns which is resembled in the current study too. Maximum mean PM_10_ and TSP of 33.42 kg 1000-hd^−1^ d^−1^ and 54.03 kg 1000-hd^−1^ d^−1^ EFs respectively were estimated in the station of the feedlot (Table 3). Considering a 24-h sampling by the BAM sampler, station 1 PM_2.5_ EF of 6.53 kg 1000-hd^−1^ d^−1^ would probably be more accepted as daily EF. However, the 10-h sampling mean EFs at station 2 and station 3 did not differ significantly either (*p* < 0.05). In the present study feedlot, the PM_10_ EF was found homogenous with a large cattle feedlot in Kansas for the 2 years period [28] and some previously published studies [9,40]. Appendix A provide previous study benchmarking EF from free-stall dairies and feedlots respectively with the current study.

Since the study also focused on the utilization of a simple emission measurement technique, the overall EF for collocation of the FRM sampler and low-cost samplers is presented in Figure 7 for comparison. The box plots’ mean PM emissions were always higher than the drone AEROCET values for both the dairy and the feedlot. The developed conversion factor for the AEROCET and drone AEROCET data may have applicability to some extent to solve this difference. Moreover, the mean Feedlot PM_2.5_ EF was close to the 90th percentile due to positive skew from the variations in the diurnal emissions. TSP and PM_10_ mean EF for feedlot was close to their median values attributing to the somewhat normal distribution. The drone AEROCET EF for both dairy and feedlot showed a slightly positively skewed distribution. The dairy EF for all PM emissions was positively skewed. However, the trend for overall mean EF distribution may attribute to the data accuracy. 

As PM emission measurement delivers emission status, it simultaneously warrants taking necessary preventive actions. The concern of controlling PM emissions from dairies and feedlots was addressed previously through airflow controlling, water sprinkles application, and management practices [41,42,43,44], of which, airflow controlling is best applied to the dairy barns and may not be applicable for the open feedlots. Pen surface treatment for moisture control with water sprinkle and mud scrapping are the most commonly recommended methods for feedlots. Since there are limitations of water resources in the High Plains where most feedlots are located, potential alternative options need to be studied for PM emission control. It is worth noting that gaseous pollutants, i.e., NH_3_, hydrogen sulfide, and VOC along with GHGs are also largely emitted from the CAFOs alongside PM, causing air quality issues in the atmosphere [45]. Reasonably, it is important to conduct more experiments on the gaseous pollutants using advance technologies for simpler emission monitoring and quicker control actions. Using gas sensors for pollutant concentration measurement and reverse dispersion modeling for emission rate prediction, emission factor can be developed for sulfur dioxide, carbon monoxide, and NOx pollutants like the PMs. Emission factors derived using dispersion modeling may provide specific information (i.e., stocking density of pen area) which can be used in finding effective and imperative control strategies.

## 4. Conclusions

TSP, PM_10_, and PM_2.5_ concentrations were measured in a dairy facility in central Texas and a representative feedlot in the Texas Panhandle along with the collocation of low-cost simple emission measuring equipment. PM emission concentrations were compared with previously published studies and nighttime emission was found higher for PM_2.5_ emissions than the day emissions. Supporting the previous studies conducted on feedlots, the current study PM ratio concluded the collected particles as coarser. The present study overall EF was within the range of other published studies for US open cattle feedlots. After data screening, the overall mean EF, in kg 1000-hd^−1^ d^−1^, estimated for the PM_2.5_, PM_10_ and TSP were 0.34, 5.59, and 15.37, respectively, for the dairy and 6.47, 21.76, and 37.10, respectively, for the feedlot.

The collocated ground low-cost samplers showed a somewhat similar pattern of distribution of PM_2.5_ emissions to the FRM sampler; however, a conversion factor must be used along with consideration of technical parameters for the achievement of actual concentration. More studies in the future are necessary to minimize potential errors in the conversion factors. The developed drone samplers may not be applicable in this regard as they exhibited a high difference of PM_2.5_ concentration with the FRM sampler probably due to height difference and turbulence from drone blades. Nonetheless, further research focusing on developing tools to reduce the mentioned causes may be beneficial.

Additionally, the annual average emission from this feedlot needs to be determined by measuring the PM_10_ and PM_2.5_ concentrations in the winter season too. Further studies throughout the year are thus necessary for actual annual PM emission estimation.

## Figures and Tables

**Figure 1 ijerph-19-14090-f001:**
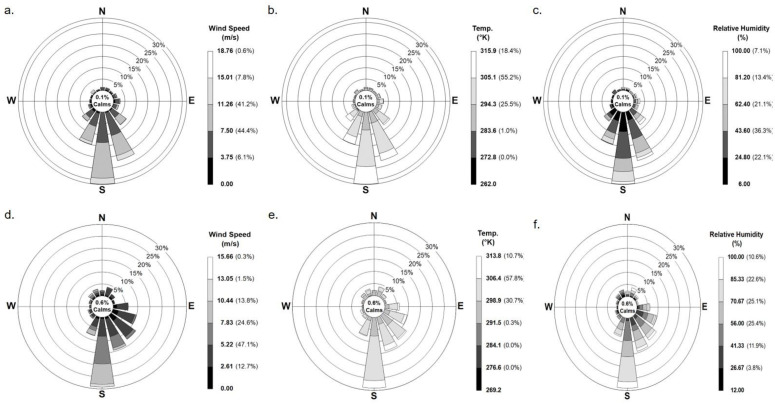
Meteorological conditions in the Feedlot C (**a**–**c**) and Dairy B (**d**–**f**) for June 2020.

**Figure 2 ijerph-19-14090-f002:**
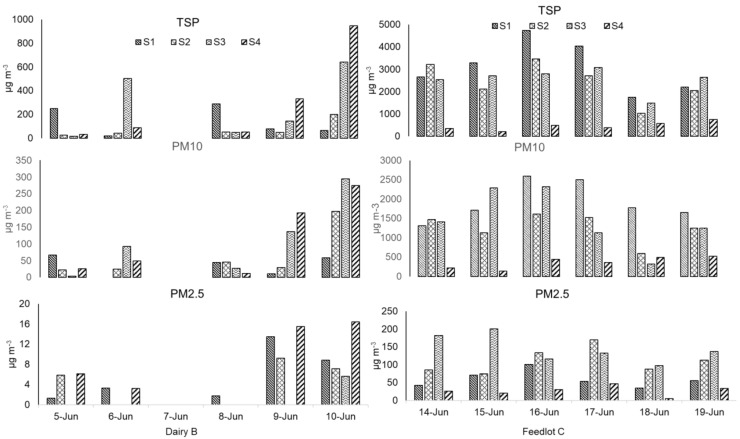
Daily 1-hr mean emission concentration (μg m^−3^) in each station for Dairy B (9 a.m. to 5 p.m.) and Feedlot C (8 a.m. to 6 p.m.).

**Figure 3 ijerph-19-14090-f003:**
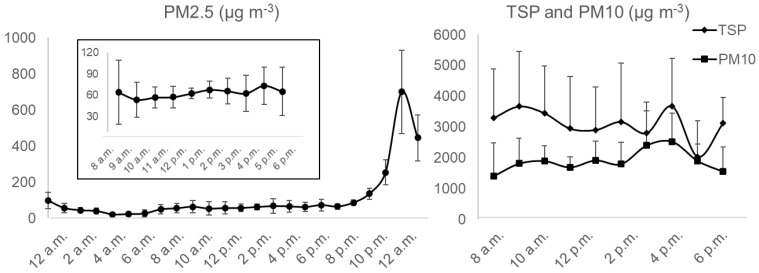
Mean diurnal TSP, PM_10_, and PM_2.5_ concentrations (μg m^−3^) during sampling hours in Station 1 of Feedlot C.

**Figure 4 ijerph-19-14090-f004:**
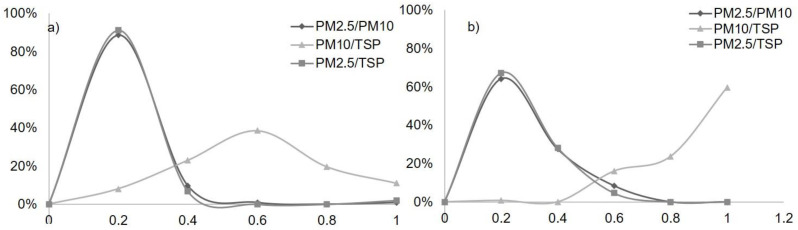
PM ratio for the collected emissions from Feedlot C (**a**) and Dairy B (**b**). (*x* and *y* axis represents ratio and % data fallen in that ratio, respectively).

**Figure 5 ijerph-19-14090-f005:**
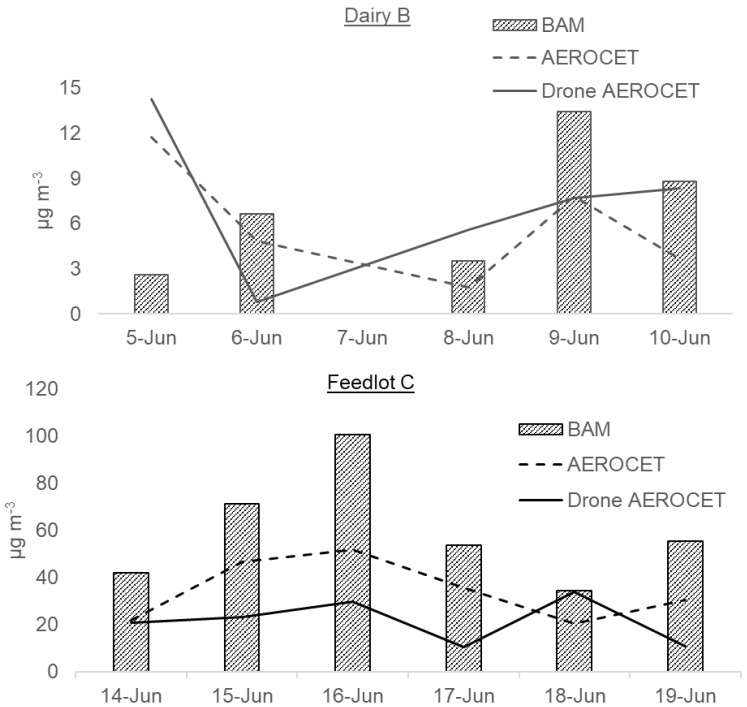
Correlation of daily means of PM_2.5_ (μg m^−3^) from BAM, AEROCET, and drone AEROCET samplers.

**Figure 6 ijerph-19-14090-f006:**
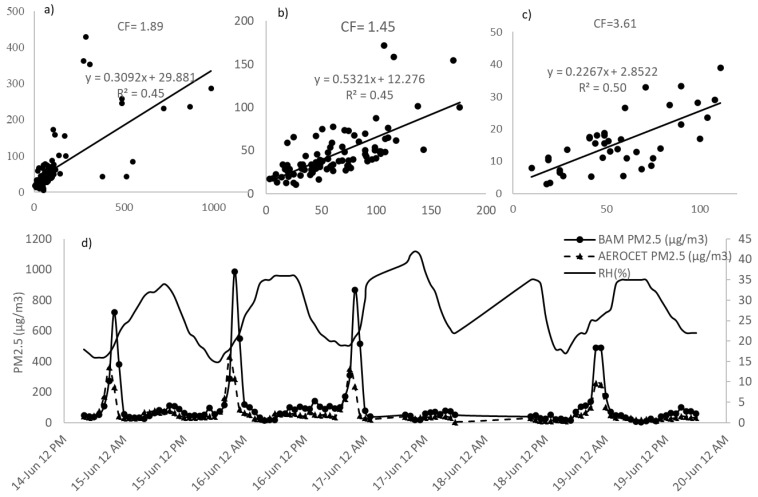
BAM PM_2.5_ emission correlation with (**a**) AEROCET samples, (**b**) outlier removed AEROCET samples, (**c**) drone AEROCET samples, and (**d**) continuous comparison of BAM and AEROCET emission from Feedlot C along with relative humidity.

**Figure 7 ijerph-19-14090-f007:**
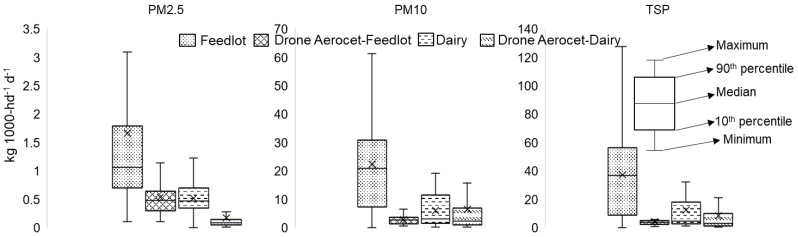
Overall collocated samplers emission factor (kg 1000-hd^−1^ d^−1^) throughout the sampling period.

**Table 1 ijerph-19-14090-t001:** Emission sampling schedule in 4 stations for both sampling sites (Station 1, 2, and 3 were in downwind position while station 4 was in upwind position).

Dairy B				Feedlot C			
Days	Station	Nominal Sampling (TSP and PM_10_) (Hr)	Nominal Sampling (PM_2.5_) (Hr)	Days	Station	Nominal Sampling (TSP and PM_10_) (Hr)	Nominal Sampling (PM_2.5_) (Hr)
5-Jun	1	6	24	14-Jun	1	10	24
	2, 3, 4	6	7		2, 3, 4	10	11
6-Jun	1	6	24	15-Jun	1	10	24
	2, 3, 4	6	7		2, 3, 4	10	11
7-Jun	1	NA	24	16-Jun	1	10	24
					2, 3, 4	10	11
8-Jun	1	8	24	17-Jun	1	10	24
	2, 3, 4	8	9		2, 3, 4	10	11
9-Jun	1	8	24	18-Jun	1	10	24
	2, 3, 4	8	9		2, 3, 4	10	11
10-Jun	1	8	24	19-Jun	1	10	24
	2, 3, 4	8	8		2, 3, 4	10	11
Lost Sample Hours *		TSP = PM_10_ = 8.33%, PM_2.5_ = 5.98%		Lost Sample Hours *		TSP = PM_10_ = 9.17%, PM_2.5_ = 5.71%	

* Data loss due to negative values, zeros, or out of range during sampling.

**Table 2 ijerph-19-14090-t002:** Mean PM_2.5_ EF (kg 1000-hd^−1^ d^−1^) in station1 FRM sampler of Dairy B and Feedlot C.

	**5-Jun**	**6-Jun**	**7-Jun**	**8-Jun**	**9-Jun**	**10-Jun**
Dairy B	0.33 * ± 0.17	0.44 ± 0.25	0.41 ± 0.21	0.46 ± 0.20	0.70 ± 0.26	0.69 ** ± 0.40
	**14-Jun**	**15-Jun**	**16-Jun**	**17-Jun**	**18-Jun**	**19-Jun**
Feedlot C	9.36 * ± 8.52	8.55 ± 5.41	7.48 ± 5.91	4.15 ± 3.56	5.14 ± 3.58	2.09 ** ± 1.62

* Based on sampling hour from 15:00 to 24:00. ** Based on sampling hour from 00:00 to 17:00.

**Table 3 ijerph-19-14090-t003:** Mean EF (kg 1000-hd^−1^ d^−1^) in each sampling stations of Dairy B and Feedlot C.

**Dairy B**				
Stations	Distance * (m)	PM_2.5_	PM_10_	TSP
S1	65 to the north	0.37 ± 0.30	2.30 ± 2.31	8.96 ± 6.91
S2	114 to the north	0.28 (single)	4.09 ± 4.31	4.70 ± 4.06
S3	35 to the north	0.14 ± 0.07	7.09 ± 6.60	17.28 ± 16.14
S4	33 to the south	0.53 ± 0.42	7.09 ± 6.66	22.07 ± 18.59
Overall		0.34 ± 0.31	5.59 ± 5.18	15.37 ± 12.38
**Feedlot C**				
Stations	Distance * (m)	PM_2.5_	PM_10_	TSP
S1	25 to the north	6.53 ± 4.48	33.42 ± 8.12	54.03 ± 17.95
S2	27 to the north	5.43 ± 4.53	21.96 ± 5.93	42.19 ± 14.10
S3	25 to the north	8.93 ± 5.92	25.23 ± 12.04	44.17 ± 8.77
S4	80 to the south **	1.15 ± 1.11	6.45 ± 2.46	8.01 ± 2.98
Overall		6.47 ± 4.12	21.76 ± 12.60	37.10 ± 21.30

* Perpendicular distance from the closest barn area. ** S4 is also 20 m far from the nearest highway.

## Data Availability

Data is contained within the article or Appendix A.

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
