# Peer review of "Particulate Matter Emission Factors for Dairy Facilities and Cattle Feedlots during Summertime in Texas"

_ijerph, 2022, doi:10.3390/ijerph192114090_

Round 1
Reviewer 1 Report
The paper is very difficult to read because it does not separate the important from the unimportant. The presentation could be improved considerably through figures that show the experimental setup. The description in Section 2.2 is very difficult to follow without a figure showing the samplers relative to area with PM emissions.
AERMOD is used to estimate PM emissions by fitting model estimates to corresponding measurements. It is not clear that the authors have an understanding of the model. In line 239, they imply that AERMOD does not use a Bi-Gaussian distribution; it actually does for unstable conditions. In 239, the authors say that dispersion causes turbulence, when it is turbulence that is responsible for dispersion.
In order to apply AERMOD to estimate emissions, the authors need to show that the model provides a good estimate of the measured PM concentrations: the concentrations at the three downwind samplers need to correlate well with model estimates using the inferred emission rate. The deviations between model estimates and measured concentrations need to be accounted for in the emission estimates. See https://doi.org/10.1016/j.agrformet.2020.108011 for an example.
The authors suggest that the low-cost AEROCET sampler can be used as a substitute for BAM to obtain quick estimates of emissions. Figures 5 and 6, which are very difficult to interpret, seem to indicate that the correspondence between the two samplers is poor. The authors need to show how the large deviations between the measurements from the low-cost and FEM instruments translate into uncertainty in emission estimates.
The paper needs a major rewrite highlighting the important aspects of the research. New figures need to be added and existing figures need to be improved in quality. It appears that the authors need to delve deeper into the formulation of AERMOD before they apply the model to estimate emissions. It is quite possible that the authors have all the data to perform a more defensible analysis.
Author Response
Thank you for your deep point outs regarding the paper. I have made a lot of changes in response in the manuscript. Also, please see my response on your comments in the attached doc file colored in blue. Thank you.

Reviewer 2 Report
This study investigated and compared the PM emission at a dairy facility and a cattle feedlot in Texas in June. I am most concerned about the authors’ usage of certain terms. For example, emission rate and concentration are two different metrics, but the authors seemed to use “emission” and “concentration” interchangeably in the manuscript. Some other terms (e.g. AERMET, AEROCET, AEROCETS, AEROCETs, Aerocet) were not properly defined, or were used in an inconsistent manner. I'm not sure why the discussion section was missing (Line 577-581). In addition, the review of previous publications is insufficient. I recommend the authors add more references, especially in the introduction section, to help the readers better understand the context of the current study. The language can also be improved, and the typos can be corrected. Some specific comments are listed below.
Line 18 and elsewhere: please put “2.5” and “10” in subscript for PM2.5 and PM10.
Line 44-46, 48-49: can the author add some references here?
Line 64: “VOCs and gases…” Gases already include VOCs.
Line 75-77: can the authors add references here?
Line 225: please define AERMET and add references.
Line 265: is “emission concentration” just the ambient PM concentration? If not, please clarify; if so, I suggest avoiding the confusing term “emission concentration” in the manuscript.
Fig 2: how do the authors explain the day-to-day variations? Is it due to the activities of the studied dairy facility and cattle feedlot? Can the authors justify that one week of sampling can sufficiently represent the PM concentrations at the sites?
Line 400-402: besides the percentages, can the authors also include the exact number of data points that were used?
Line 495: the abbreviation EF was defined in Line 275, again in Line 284, and again here.
Line 577-581: is the discussion section missing?
Author Response
Thank you. Please see the attached response to your comments. I have made the changes suggested. Thank you.

Round 2
Reviewer 2 Report
NA